# Aflatoxin Mixture-Driven Intrahepatic Cholangiocarcinoma in Rats Involving G1/S Checkpoint Dysregulation

**DOI:** 10.3390/toxins18010014

**Published:** 2025-12-25

**Authors:** Vinícius Menezes Braga, Paulo Henrique Fernandes Pereira, Letícia de Araujo Apolinario, Deisy Mara Silva Longo, Leandra Naira Zambelli Ramalho, Sher Ali, Carlos Augusto Fernandes de Oliveira, Fernando Silva Ramalho

**Affiliations:** 1Department of Pathology and Legal Medicine, Ribeirão Preto Medical School, University of São Paulo, Av. Bandeirantes, 3900, Ribeirão Preto 14049-900, SP, Brazil; vinicius.menbraga@usp.br (V.M.B.); paulohf.pereira@usp.br (P.H.F.P.); dms@fmrp.usp.br (D.M.S.L.); lramalho@fmrp.usp.br (L.N.Z.R.); 2Department of Nursing in Hospital Care, Federal University of the Triângulo Mineiro, R. Vigário Carlos, 100, Uberaba 38025-350, MG, Brazil; leticia.apolinario@uftm.edu.br; 3Department of Food Engineering, School of Animal Science and Food Engineering, University of São Paulo, Av. Duque de Caxias Norte, 225, Pirassununga 13635-900, SP, Brazil; alisher@usp.br

**Keywords:** aflatoxins, AFB_1_-dominant aflatoxin mixture, intrahepatic cholangiocarcinoma, immunohistochemistry, TP53, cyclin D1, retinoblastoma protein, PCNA, β-catenin

## Abstract

Aflatoxins (AFs) are potent hepatotropic mycotoxins—AFB_1_ being the best-characterized—yet their ability to induce intrahepatic cholangiocarcinoma (iCCA) remains underexplored. Male Wistar rats received vehicle (controls; *n* = 5) or an AFB_1_-dominant AF mixture (AFB_1_ 39.46 μg/mL; AFB_2_ 1.13 μg/mL; AFG_1_ 17.44 μg/mL; AFG_2_ 0.59 μg/mL— *n* = 10) by daily gavage for 90 days, at a dose equivalent to 400 μg AFB_1_ per kg of diet. After 12 months, twelve iCCA tumors were resected and analyzed by histology (H&E) and tissue-microarray-based immunohistochemistry (Cytokeratin-19, Hep Par-1, p53, Cyclin D1, Rb, β-catenin, and PCNA). Lesions predominantly showed glandular/tubular architecture consistent with iCCA and were cytokeratin-19-positive and Hep Par-1-negative. Cell proliferation was high (PCNA ≈ 69%). p53 displayed nuclear accumulation in 83% of tumors. G1/S control was perturbed, with cyclin D1 overexpression (67%), and Rb was positive in 58% of iCCA. Aberrant Wnt activation was rare (nuclear β-catenin in 8%). Subchronic exposure to an AFB_1_-dominant AF mixture in rats was associated with iCCA characterized by high proliferative activity, p53 accumulation, and disruption of G1/S checkpoint components. These findings broaden the oncogenic spectrum of AFs and warrant genomic confirmation of AF mutational signatures.

## 1. Introduction

Intrahepatic cholangiocarcinoma (iCCA) is a malignant neoplasm arising from the intraparenchymal biliary epithelium and accounts for roughly 20% of primary liver cancers, with substantially worse survival than many hepatocellular tumors. Classic iCCA risk factors include chronic cholangitis, hepatobiliary trematodes, and specific toxic exposures; however, aflatoxins (AFs) are seldom considered in this etiology [1,2]. AFs are a family of mycotoxins produced by *Aspergillus* species. Among them, aflatoxin B_1_ (AFB_1_) is the best characterized and most potent. Inadequate food storage favors fungal growth and dietary contamination, creating exposure scenarios of public-health relevance [3,4,5].

Mechanistically, AF toxicity, classically modeled for AFB_1_, results from hepatic bioactivation to the aflatoxin-8,9-epoxide, which forms mutagenic DNA adducts (e.g., AFB_1_-N^7^-guanine). The resultant mutational spectrum includes the emblematic G → T transversion at TP53 codon 249 (R249S), a molecular fingerprint of AF exposure [6,7]. Loss of p53 function permits replication of damaged DNA, fostering genomic instability and aggressive behavior [5,6,7].

Although AFs are historically linked to hepatocarcinogenesis, experimental evidence suggests they may also induce lesions with cholangiocarcinoma-like features. A biologically plausible route is biliary excretion of genotoxic metabolites, which chronically exposes cholangiocytes to mutagenic stress. In addition, bipotent hepatic progenitors (“oval cells”) at the hepatocyte-cholangiocyte interface can expand under diffuse toxic injury and, when subjected to aflatoxin-induced DNA damage, undergo malignant transformation. Because morphology alone may be insufficient in poorly differentiated tumors, immunohistochemistry (IHC) is pivotal for lineage assignment [Cytokeratin 19 (CK19) for biliary/progenitor, Hepatocyte Paraffin 1 (Hep Par-1) for hepatocellular] and for interrogating oncogenic pathways (p53, Wnt/β-catenin) and G1/S checkpoint components (cyclin D1/CDK4 and the Rb/E2F axis). Proliferating cell nuclear antigen (PCNA) further quantifies proliferative activity. Thus, this study aimed to evaluate whether subchronic exposure to an AFB_1_-dominant AF mixture can induce iCCA in rats and to characterize the resulting tumors through histology and immunohistochemistry, focusing on lineage markers, proliferation, and key oncogenic pathways.

## 2. Results

### 2.1. General Lesion Features

Eight of 10 rats exposed to the AFB_1_-dominant AF mixture developed macroscopically visible hepatic tumors (none in controls). The appearance of the nodular lesions can be observed in Figure 1. Tumors per animal ranged from 0 to 3 (median 1). Histologically, we identified 12 malignant epithelial lesions: 11 iCCAs and 1 combined hepatocellular carcinoma–cholangiocarcinoma (cHCC-CCA). Thus, iCCA-phenotype tumors (including the cHCC-CCA case) occurred in 8/10 exposed rats (80%; 12 tumors in total; median 1 tumor per affected animal, range 0–3), compared with 0/5 controls (0/5 rats with tumors).

### 2.2. Morphology

In the 12 cases, the predominant architecture comprised irregular glands, ducts, and tubulo-acinar structures lined by atypical epithelial cells with desmoplastic stroma—features consistent with iCCA (Figure 2A,B). These observations indicate biliary-type tumors in this experimental context.

### 2.3. Lineage Markers—CK19 and Hep Par-1

Immunohistochemistry supported biliary differentiation in most tumors. CK19 showed diffuse membranous/cytoplasmic positivity in all evaluable samples (Figure 2C–F), whereas Hep Par-1 immunoreactivity was rare. Morphologically, iCCA lesions predominantly displayed well-formed glandular/tubular structures lined by cuboidal-to-columnar epithelium with enlarged hyperchromatic nuclei, prominent nucleoli, desmoplastic stroma, and occasional intraluminal mucin. One tumor was a cHCC-CCA, with bile-duct-like glands adjacent to trabecular and pseudoacinar hepatocellular plates; the hepatocellular component stained positive for Hep Par-1 (Figure 2G), while the biliary component was CK19-positive (Figure 2H). Immunohistochemical evaluation of normal livers (controls) and non-tumorous hepatic parenchyma showed physiological baseline staining patterns, such as CK19 restricted to native bile ducts, and Hep Par-1 strongly expressed in normal hepatocytes.

All tumors were evaluated by both morphology and a standardized immunohistochemical panel (CK19, Hep Par-1, PCNA, p53, cyclin D1, Rb, and β-catenin), ensuring consistent lineage assignment and improving diagnostic confidence. Although several lesions displayed classic tubular/glandular features compatible with iCCA, no tumor was classified on morphology alone. Immunophenotyping was systematically applied to confirm biliary differentiation and exclude hepatocellular lineage, thereby increasing diagnostic reliability. After integrating histologic and IHC findings, 11 tumors were classified as iCCA and one tumor as cHCC-CCA.

### 2.4. Proliferative Index (PCNA)

Neoplasms exhibited markedly increased proliferative activity compared with non-tumorous liver. PCNA demonstrated diffuse, strong nuclear labeling in most samples, with a mean proliferation index of ~69% positive nuclei (range ~50–80%, individual values are detailed in Table 1). Control livers and non-tumorous parenchyma displayed <5% PCNA-positive nuclei, limited to periportal regenerative zones. Morphology and lineage markers are shown in Figure 2.

### 2.5. Oncogenic Pathways

#### 2.5.1. p53 Expression

In 10/12 tumors (83%), there was intense, widespread nuclear p53 labeling—occasionally with cytoplasmic deposition—consistent with abnormal protein stabilization and compatible with, but not specific for, TP53 pathway alterations. The remaining ~17% showed absent or sparse labeling. Several CK19-positive tumors exhibited strong p53 accumulation, indicating that TP53 alterations occur in biliary-lineage lesions.

#### 2.5.2. Wnt/β-Catenin Pathway

β-Catenin labeling was predominantly membranous/cytoplasmic in 11/12 lesions, similar to normal liver, indicating lack of aberrant Wnt activation in most tumors. Only 1/12 (8%) displayed nuclear β-catenin. Collectively, these data suggest that Wnt/β-catenin activation is uncommon in this model.

#### 2.5.3. Cell-Cycle Markers—Cyclin D1 and Rb

Cyclin D1 showed nuclear overexpression in 8/12 tumors (67%), with 5–30% of tumor nuclei positive. Rb expression was lost in 5/12 (42%), whereas 7/12 (58%) retained nuclear Rb. Notably, cyclin D1-positive tumors with either Rb loss or Rb retention indicate convergent mechanisms of G1/S deregulation, via Rb deficiency and/or CDK4/6-mediated phosphorylation when Rb is present.

### 2.6. IHC Summary

Across the 12 tumors, marker frequencies were: CK19, 12/12 (100%); p53, 10/12 (83%); nuclear β-catenin, 1/12 (8%); cyclin D1, 8/12 (67%); and Rb loss, 5/12 (42%). Approximately 83% (10/12) of neoplasms fulfilled our IHC criteria suggestive of G1/S deregulation (cyclin D1↑, Rb↓, or both). Positivity cutoffs were: CK19/Hep Par-1/β-catenin ≥ 10% in the expected pattern; p53 ≥ 10% strong nuclear; cyclin D1 ≥ 10% nuclear; and Rb loss defined as 0% tumor nuclei stained in the presence of internal positive controls. The proliferation index and part of the IHC summary are presented in Figure 3.

## 3. Discussion

This study provides descriptive evidence that subchronic exposure to an AFB_1_-dominant AF mixture was associated with the development of hepatic tumors in rats with a prevailing iCCA phenotype [8,9,10], as supported by morphology and a biliary-lineage immunoprofile (CK19 positivity with rare Hep Par-1 detection) [11,12]. The integrated immunohistochemical panel further delineates recurrent pathway alterations, high proliferative activity (PCNA), frequent p53 accumulation, and G1/S checkpoint deregulation (cyclin D1 overexpression with Rb loss), while indicating that Wnt/β-catenin activation is uncommon. Taken together, the data are compatible with a working model in which AF exposure contributes to iCCA development through TP53 pathway perturbation and changes in G1 → S control, as inferred from the IHC profile [13].

Previous reports have linked AF exposure to biliary-type liver tumors in other species, but under quite different experimental conditions. In the classic study by Moore et al., male F344 rats and Syrian golden hamsters received AFB1 by oral gavage 5 days per week for 6 weeks at doses of 1 and 2 mg/kg body weight/day, respectively, with or without subsequent phenobarbital promotion; follow-up extended to 46 weeks in rats and 78 weeks in hamsters. Under these near-toxic, high-dose regimens, all 50 treated rats eventually developed hepatocellular carcinomas, many with concomitant iCCA or cHCC-CCA, whereas in hamsters, cholangiocarcinomas predominated (15/49 animals at 78 weeks) in the setting of widespread microscopic cholangiomas. However, lesion classification in that work relied on conventional histology, without systematic use of lineage markers or pathway-focused immunohistochemistry, and dose–response relationships for cholangiocellular vs. hepatocellular tumors could not be formally dissected [9]. A separate short report in a rhesus monkey described the induction of cholangiocarcinoma after AF treatment, but this observation was based on a single animal, with limited dosing details and no molecular characterization, underscoring its anecdotal nature [10]. Against this background, our rat model complements the existing literature by showing that subchronic exposure to an AFB1-dominant AF mixture for 90 days, at a dose equivalent to 400 µg AFB1 per kg of diet, can result in predominantly iCCA-phenotype tumors after 12 months, defined not only by morphology but also by a CK19^+^/Hep Par-1^−^ biliary-lineage immunoprofile and an integrated panel of proliferation and cell-cycle markers (PCNA, p53, cyclin D1, Rb, β-catenin). While our cohort size and descriptive design preclude formal incidence or dose–response modeling, the combination of controlled subchronic exposure, systematic histologic review, and pathway-oriented immunophenotyping refines the characterization of aflatoxin-associated cholangiocarcinogenesis and provides a framework for future mechanistic and genomic studies.

In terms of real-world relevance, this regimen falls within the range of high-dose experimental hepatocarcinogenesis models rather than typical human dietary exposure. While the dose likely exceeds most chronic human scenarios, it approximates the upper end of exposures reported in highly contaminated settings and is consistent with previously reported high-dose experimental hepatocarcinogenesis regimens in rodents [9].

The biological plausibility that AFs initiate cholangiocarcinogenesis arises from hepatocellular bioactivation followed by biliary excretion of genotoxic metabolites, which chronically exposes the ductal epithelium to mutagenic stress [3,4]. In parallel, expansion of bipotent hepatic progenitors (“oval cells”) under diffuse toxic injury provides a susceptible compartment in which aflatoxin-induced DNA damage may fixate, yielding malignant clones capable of biliary differentiation. The predominance of CK19 positivity and the absence of Hep Par-1 in most lesions favor cholangioepithelial origin; tumors lacking both markers likely reflect very poor differentiation or technical loss of lineage antigens rather than alternative lineage commitment [14].

Diffuse nuclear p53 accumulation in most tumors (10/12, 83%) supports abnormal protein stabilization compatible with TP53 alterations, an observation aligned with the canonical AF mutational footprint [6,7]. Functionally, loss of p53 eliminates a key barrier to cell-cycle arrest and apoptosis in the face of DNA adducts, allowing clonal expansion under genotoxic pressure. This framework is consistent with the high proliferative fraction observed and provides a mechanistic bridge to checkpoint deregulation. Importantly, nuclear accumulation of p53 by IHC is not synonymous with TP53 mutation and may also result from wild-type protein stabilization under cellular stress; therefore, sequencing would be required to demonstrate the presence and spectrum of TP53 mutations in this model [15].

Interestingly, one tumor displayed a cHCC-CCA phenotype [16,17], showing adjacent trabecular/pseudoacinar hepatocellular plates and well-formed ductular structures. Immunohistochemically, the hepatocellular component was positive for Hep Par-1 and negative for cytokeratin 19 (CK19), while the biliary component demonstrated the inverse profile (CK19^+^/Hep Par-1^−^) [11], confirming true biphenotypic differentiation rather than sampling heterogeneity.

Both compartments exhibited strong nuclear p53 accumulation, suggesting a common clonal origin with shared DNA damage response alterations [18,19]. Notably, loss of RB protein expression was restricted to the biliary component, whereas the hepatocellular areas retained nuclear RB staining. This divergence indicates that G1/S checkpoint disruption might preferentially contribute to the progression of the cholangiocellular lineage within mixed lesions, highlighting intratumoral heterogeneity [19] and lineage-specific vulnerabilities to aflatoxin-induced oncogenic stress. Similar biphenotypic patterns and differential expression of CK19, HepPar-1, and cell-cycle regulators (p53, Rb) have been previously documented in human cHCC-CCA [20], supporting the notion of a monoclonal origin with lineage divergence during tumor evolution.

Overexpression of cyclin D1 in 8/12 tumors (67%), coupled with Rb loss in 5/12 (42%), is compatible with convergent acceleration of G1 → S transition. The co-occurrence of cyclin D1 positivity within the Rb-loss group is biologically coherent with cyclin D1/CDK4/6-mediated Rb inactivation [21]. From a translational perspective, our IHC profile raises the hypothesis that the CDK4/6 axis could be explored as a therapeutic target in aflatoxin-associated biliary tumors, enabling new targeted-therapy approaches as well as other molecular strategies, including FGFR2-directed inhibitors, IDH1/2 inhibitors, ERBB2-targeted therapies and selected immunotherapy-based combination [22], and besides that cyclin D1 and Rb status might represent markers of progression or aggressiveness. These possibilities remain speculative and require dedicated experimental testing. This interpretation aligns with current reviews showing that cholangiocarcinoma exhibits frequent dysregulation of cell-cycle and proliferative pathways, reinforcing the rationale for exploring CDK4/6 axis inhibition as a potential therapeutic strategy. Accordingly, Rb loss in our series should be interpreted as absence of detectable protein expression rather than proof of RB1 mutation; epigenetic silencing, altered transcription, or functional inactivation through hyperphosphorylation cannot be distinguished on the basis of IHC alone [23].

Predominantly membranous/cytoplasmic β-catenin with rare nuclear localization indicates that aberrant Wnt activation is not a principal driver in this model. This is concordant with many iCCA molecular landscapes—where Wnt pathway activation is less prominent than in several hepatocellular contexts—and reinforces the centrality of TP53 and cell-cycle lesions under AF exposure [24]. In contrast to hepatocellular carcinoma, where CTNNB1 mutations define a major Wnt-activated molecular subclass, large-scale genomic studies indicate that iCCA is primarily driven by IDH1/2, FGFR2, BAP1 and chromatin-remodeling lesions, with CTNNB1 alterations being rare or virtually absent [25,26,27,28,29]. Our finding of predominantly membranous β-catenin staining with only rare nuclear accumulation therefore aligns with the notion that canonical Wnt/β-catenin activation is not a common oncogenic mechanism in iCCA.

Lineage assignment relied on complementary markers with orthogonal biology (CK19 for cholangiocytes/progenitor, Hep Par-1 for mature hepatocytes), decreasing the risk of single-marker misclassification. Nevertheless, immunohistochemistry infers function indirectly: Rb “loss” by IHC operationalizes absence of detectable protein in the presence of internal controls but does not discriminate among truncating mutations, epigenetic silencing, or hyperphosphorylation-mediated functional inactivation. Similarly, p53 accumulation suggests but does not prove mutation; sequencing is required for definitive attribution.

Although the data presented here support that subchronic exposure to an AFB_1_-dominant AF mixture in rats can induce iCCA-phenotype hepatic tumors under the conditions tested, this work has limitations. The cohort size (10 exposed animals) and descriptive design limit incidence-level inferences and preclude robust dose–response analyses. Only young males were evaluated; sex- or age-dependent susceptibility could differ. Because of the small cohort size and the exploratory nature of the study, we restricted our analysis to descriptive statistics and did not perform formal hypothesis testing. This limits the strength of any incidence estimates and precludes dose–response or risk modeling inferences.

Establishing that an AFB_1_-dominant AF mixture can produce iCCA-phenotype tumors in rats widens the recognized spectrum of aflatoxin-related oncogenesis beyond hepatocellular contexts and underscores a biliary hazard in chronically exposed settings. Next steps should include: (i) genomic validation (whole-exome/targeted sequencing to confirm SBS24 and define co-mutational patterns); (ii) functional interrogation of the CDK4/6–Rb axis (pharmacologic inhibition, phospho-Rb readouts, cell-cycle modeling); (iii) assessment of p53 reactivation strategies or synthetic-lethal dependencies in DNA-damage response; and (iv) cofactor models (e.g., chronic cholangitis or viral hepatitis) to probe environmental synergy and lineage bias. Cross-validation in human cohorts from high-aflatoxin regions, pairing immunophenotypic profiles with mutational signatures, could consolidate the translational relevance.

Briefly, the data are compatible with a mechanistic model in which biliary exposure to AF metabolites, failure of TP53 safeguards, and G1/S checkpoint deregulation act in concert to support cholangiocarcinogenesis. This integrated view offers supportive leads for subsequent molecular confirmation and exploratory therapeutic studies in toxin-driven iCCA.

## 4. Conclusions

Subchronic exposure to an AFB_1_-dominant AF mixture in rats produced hepatic tumors with a predominantly intrahepatic cholangiocarcinoma (iCCA) phenotype, as indicated by morphology and a biliary-lineage immunoprofile (CK19 positivity. An integrated immunohistochemical panel revealed high proliferative activity (PCNA), frequent p53 accumulation consistent with TP53 pathway disruption, and G1/S checkpoint deregulation marked by cyclin D1 overexpression coupled with Rb loss, whereas Wnt/β-catenin activation was uncommon. Together, these findings delineate a toxin-driven iCCA axis centered on failure of genomic safeguards and accelerated G1 → S progression.

Mechanistically, the data are consistent with a model in which biliary excretion of genotoxic AF metabolites sustains cholangiocyte exposure, fostering DNA damage fixation and clonal expansion. Beyond supporting an iCCA-phenotype model in rats, the work highlights candidate biological nodes—notably the CDK4/6–Rb axis and p53-associated vulnerabilities—that warrant hypothesis-generating functional and pharmacologic studies. The convergent histopathologic and immunophenotypic evidence supports the etiologic role of an AFB1-dominant AF mixture in cholangiocarcinogenesis and broadens the recognized oncogenic spectrum of AFs.

## 5. Materials and Methods

### 5.1. Ethical Approval and Regulatory Compliance

All procedures involving animals complied with national regulations and ARRIVE 2.0 [30] guidelines and were approved by the Institutional Animal Care and Use Committee (CEUA) of Ribeirão Preto Medical School, University of São Paulo, Brazil (protocol 127/13; approval received on 27 January 2014).

### 5.2. Animals and Housing Condition

Male Wistar rats (*n* = 10 exposed; *n* = 5 vehicle controls), ~150 g and 8 weeks old at study start, were obtained from a dedicated specific-pathogen-free facility. Animals were acclimatized for 7 days prior to dosing and maintained under controlled conditions (22 ± 2 °C; 45–65% humidity; 12:12 h light–dark cycle) with ad libitum access to standard chow and filtered water. Health status was checked daily by trained personnel.

### 5.3. Aflatoxin Mixture and Dosing Regimen

Rats in the exposed group received an AFB_1_-dominant AF mixture by oral gavage once daily for 90 consecutive days. The dosing preparation (diluted in vegetable oil, protected from light) contained the following concentrations (μg/mL): AFB_1_: 39.46; AFB_2_: 1.13; AFG_1_: 17.44; AFG_2_: 0.59. The control group received vehicle only under identical conditions. Dosing solutions were prepared under low-light conditions, stored at 4 °C, and equilibrated to room temperature before use. Immediately before administration, each solution was vortexed to ensure homogeneity, and then delivered by daily gavage for 90 days at a dose equivalent to 400 µg AFB_1_ per kg of diet. The doses to be administered orally to each animal took into account the individual feed consumption of the previous 3 days. Since the animals were kept in cages containing 4 rats of similar weights, daily feed consumption was evaluated collectively, and the average value was obtained from the 4 animals.

### 5.4. Experimental Design, Clinical Monitoring, and Follow-Up

Body weight and general clinical condition (activity, posture, piloerection, food/water intake) were recorded weekly during exposure and monthly thereafter. Humane endpoints (≥20% body-weight loss, severe lethargy, persistent self-neglect, refractory pain/distress) triggered immediate veterinary evaluation and euthanasia when indicated. Animals were followed for 12 months from exposure onset. At the study endpoint (or upon reaching a humane endpoint), rats were anesthetized and humanely euthanized. The body weight of the rats at euthanasia ranged from 680 to 770 g. No premature euthanasia was performed.

### 5.5. Macroscopic Liver Examination and Tissue Sampling

Complete necropsies were performed. Livers were examined macroscopically; the number, location, and largest diameter of visible nodules were recorded. Representative samples were collected from each tumor nodule and from macroscopically non-tumorous liver parenchyma. In total, 12 distinct hepatic tumors were identified among exposed animals (≈1 tumor per animal; median 1; range 0–3). No nodules were observed in controls.

### 5.6. Histopathology

Tissue fragments were fixed in 10% neutral buffered formalin (24–48 h), processed routinely, embedded in paraffin, and sectioned at 4 μm for hematoxylin–eosin (H&E). A board-certified veterinary pathologist, blinded to group allocation, evaluated architectural patterns (glandular/tubular, solid), cytologic atypia, mitotic index, necrosis, and vascular invasion. Glandular/tubular structures lined by cuboidal/columnar cells with desmoplastic stroma were interpreted as consistent with iCCA.

### 5.7. Immunohistochemistry and Scoring Criteria

Formalin-fixed, paraffin-embedded tissue specimens were sectioned at 4 µm thickness, mounted on silanized glass slides, and dried overnight at 37 °C. Immunohistochemical staining was performed using the LSAB+ System–HRP (Dako, Glostrup, Denmark) according to the manufacturer’s instructions, with minor adjustments as specified below.

Sections were deparaffinized in xylene and rehydrated through graded ethanol to distilled water, followed by heat-induced antigen retrieval in 10 mmol L^−1^ of citrate buffer (pH 6.0) for 45 min. After this period, slides were then allowed to cool to room temperature (25 °C) and rinsed in Tris-buffered saline solution. Endogenous peroxidase activity was quenched by incubation with methanol: hydrogen peroxide (97:3, *v*/*v*) for 5 min, and nonspecific binding was minimized using the protein blocking solution provided in the LSAB+ Dako System-HRP.

Sections were incubated overnight at 4 °C with primary antibodies as follows: CK19 (clone b170, 1:100 *v*/*v*, Novocastra, Newcastle upon Tyne, UK); Hep Par-1/HSA (clone OCH1E5, 1:400 *v*/*v*, Cell Marque, Rocklin, CA, USA); p53 (clone Pab 1801, 1:100 *v*/*v*, Santa Cruz Biotechnology, Dallas, TX, USA); Cyclin D1 (clone 92G2, 1:50 *v*/*v*, Cell Signaling, Danvers, MA, USA); Rb (ab85607, 1:200 *v*/*v*, Abcam, Cambridge, MA, USA); β-catenin (clone D10A8, 1:100 *v*/*v*, Cell Signaling); PCNA (clone PC10, 1:100, Santa Cruz Biotechnology).

After this period, the slides were washed in phosphate-buffered saline and treated with biotinylated universal secondary antibody (LSAB+ Dako kit) for 30 min. Streptavidin–horseradish peroxidase (LSAB+ Dako kit) was subsequently applied, after which immunoreactivity was developed with 3,3′-diaminobenzidine tetrahydrochloride (DAB chromogen, Thermo Fisher Scientific, Waltham, MA, USA) in phosphate-buffered saline with 0.036% hydrogen peroxide (pH 7.5) for 5 min. Slides were counterstained with Mayer’s hematoxylin (Thermo Fisher Scientific), dehydrated in graded ethanol, cleared in xylene, and mounted with Permount (Thermo Fisher Scientific).

For each run, internal positive controls (bile ducts for CK19; non-neoplastic hepatocytes for Hep Par-1 and Rb nuclear pattern; crypt epithelium/tonsil for cyclin D1; hepatocyte membrane for β-catenin) and reagent blanks (primary antibody omitted) were included.

Lineage assignment and pathway status were assessed by an experienced pathologist who was blinded to all experimental data. Protein expression was classified as positive when the proportion of tumor cells exhibiting immunoreactivity exceeded the predefined cutoff value. Table 2 summarizes the immunohistochemical expression patterns, cutoff values, and the corresponding biological interpretation for all proteins evaluated.

Additionally, the intensity of tumor cell replication was measured through the nuclear expression of PCNA. The cell proliferation index was determined by counting 500 tumor cells across five high-power fields (hot-spots), expressing the fraction (%) of PCNA-positive nuclei.

### 5.8. Statistical Approach

Given the exploratory scope and cohort size, analyses were descriptive, reporting proportions and ranges. No formal hypothesis testing or sample-size power calculations were performed. All antibodies and detection reagents were used within the manufacturer-recommended shelf life and stored as specified. Fresh DAB was prepared for each staining run. Batch-to-batch comparability was monitored by (i) constant inclusion of the same internal control tissues; (ii) review of background staining; and (iii) run acceptance only when internal controls met pre-defined criteria (expected pattern/intensity present).

## Figures and Tables

**Figure 1 toxins-18-00014-f001:**
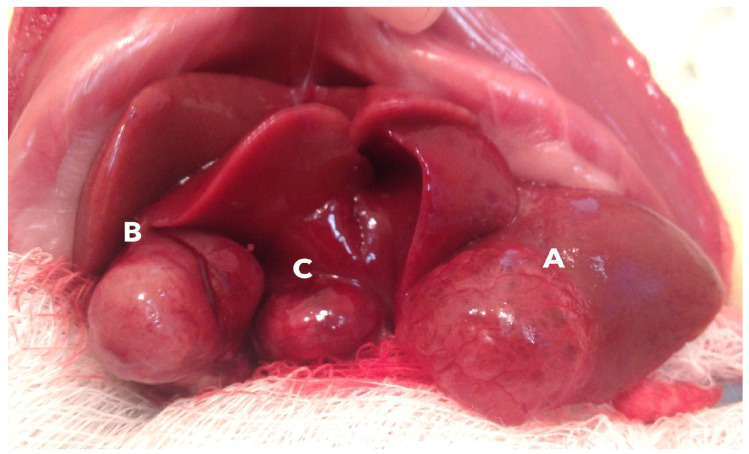
Macroscopic aspect of the rat liver showing cystic tumor nodules in the left lateral lobe (1.76 cm^3^—A), right lateral lobe (3.78 cm^3^—B) and caudate lobe (1.22 cm^3^—C). The values in parentheses correspond to the estimated tumor volumes.

**Figure 2 toxins-18-00014-f002:**
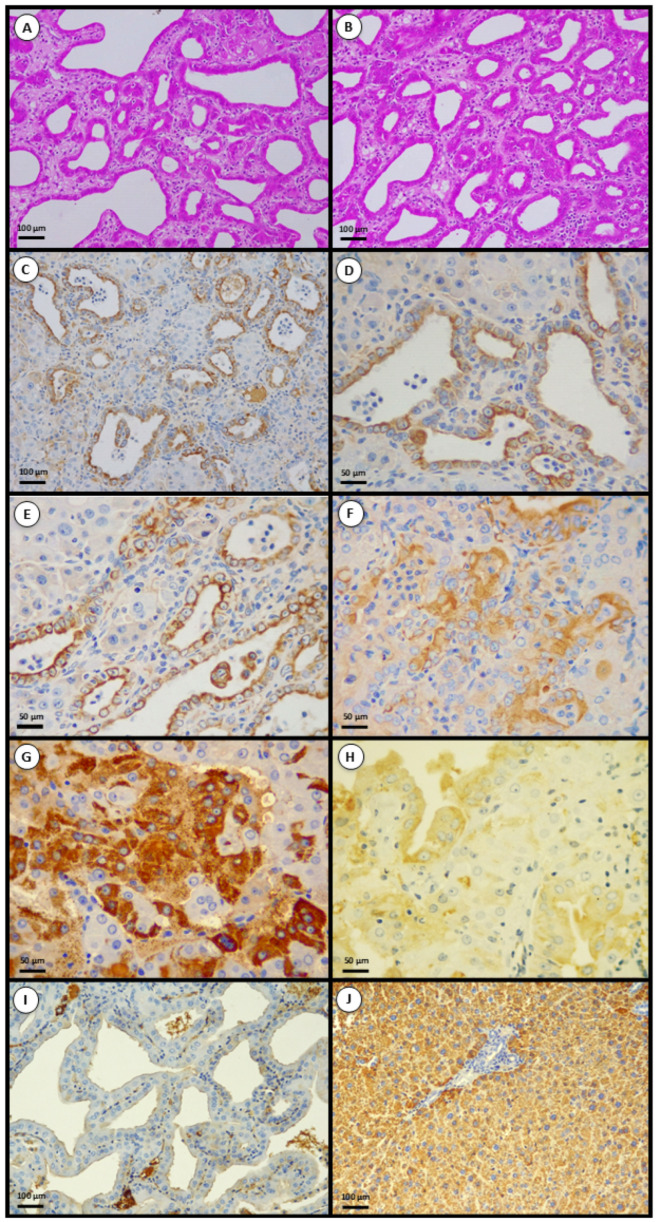
Representative photomicrographs of hepatic tumors induced by an AFB1-dominant aflatoxin mixture. (**A**,**B**) Markedly disorganized hepatic lobular architecture with multiple tubular and glandular structures lined by tumor cells displaying large, pleomorphic, hyperchromatic nuclei and desmoplastic stroma, consistent with intrahepatic cholangiocarcinoma (H&E). (**C**–**F**) Neoplastic cells exhibiting strong cytoplasmic expression of cytokeratin-19 (CK19), a standard marker of cholangiocytes, confirming biliary differentiation. Combined hepatocellular carcinoma–cholangiocarcinoma showing Hep Par-1 expression restricted to the hepatocellular component (**G**), and cytoplasmic CK19 positivity in the cholangiocarcinoma component of the same lesion (**H**). Tumoral biliary epithelium lacking Hep Par-1 immunolabeling (**I**), in contrast to normal hepatocytes exhibiting diffuse cytoplasmic Hep Par-1 positivity (**J**). Scale bars: [100 μm (**A**–**C**,**I**,**J**); 50 μm (**D**–**H**)].

**Figure 3 toxins-18-00014-f003:**
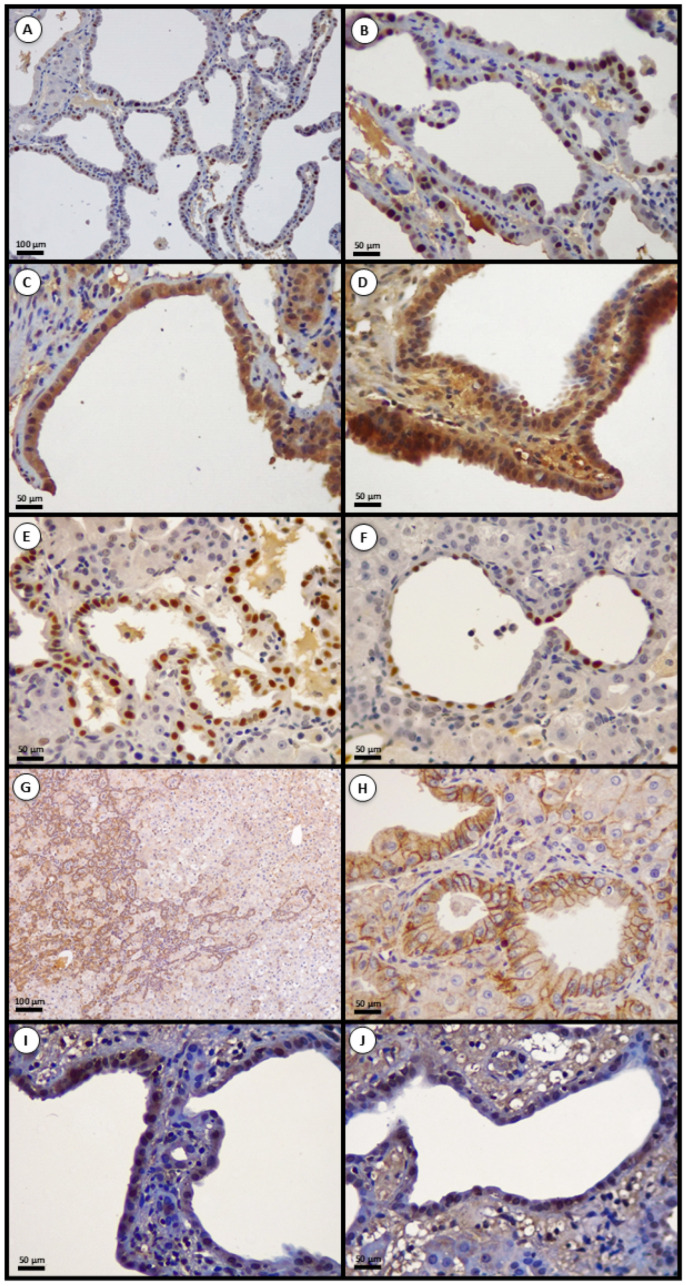
Representative immunohistochemical profiles of aflatoxin-associated intrahepatic cholangiocarcinomas. (**A**,**B**) Positive nuclear expression of proliferating cell nuclear antigen (PCNA) in tumor cells, indicating increased proliferative activity. (**C**,**D**) Tumor samples showing strong nuclear and cytoplasmic immunostaining for p53 protein, consistent with abnormal protein stabilization. (**E**,**F**) Intense nuclear immunostaining for cyclin D1 in neoplastic cells. (**G**,**H**) Predominantly membranous/cytoplasmic β-catenin in tumor cells, consistent with lack of aberrant Wnt/β-catenin activation. (**I**,**J**) Tumor cells showing nuclear expression of retinoblastoma protein (Rb). Panels illustrate typical patterns for the majority of tumors with the indicated marker status. Scale bars: [100 μm (**A**,**G**); 50 μm (**B**–**F**,**H**–**J**)].

**Table 1 toxins-18-00014-t001:** Immunohistochemical panel and classification of tumors. CK19 and Hep Par-1 were used as gold-standard markers of biliary ducts and hepatocytes, respectively. PCNA highlights cell proliferation rate, while p53, cyclin D1, Rb and β-catenin were applied as G1/S checkpoint components. No aberrant or neoplastic-like staining was identified in liver samples of control rats—animals 11, 12, 13, 14 and 15.

*Animal*	*Tumor*	*CK19*	*Hep Par-1*	*Classification*	*PCNA*	*p53*	*Cyclin D1*	*Rb*	*β-* *Catenin*
** *1* **	**A**	✅	❌	**iCCA**	66.3%	✅	✅	❌	❌
** *2* **	**B**	✅	❌	**iCCA**	55.5%	✅	✅	❌	❌
** *3* **	**C**	✅	❌	**iCCA**	72.1%	✅	❌	✅	❌
** *3* **	**D**	✅	❌	**iCCA**	80.2%	✅	✅	✅	❌
** *4* **	**E**	✅	❌	**iCCA**	58.9%	✅	✅	✅	❌
** *5* **	**F**	✅	❌	**iCCA**	77.5%	✅	✅	✅	❌
** *5* **	**G**	✅	❌	**iCCA**	50.9%	✅	✅	✅	❌
** *5* **	**H**	✅	❌	**iCCA**	79.5%	✅	✅	✅	❌
** *6* **	**I**	✅	❌	**iCCA**	67.3%	❌	❌	❌	❌
** *6* **	**J**	✅	❌	**iCCA**	80.8%	✅	✅	❌	✅
** *7* **	**K**	✅	❌	**iCCA**	65.7%	❌	❌	❌	❌
** *8* **	**L**	✅	✅	**cHCC-CCA**	71.2%	✅	❌	✅	❌
** *9, 10* **	—	✅	✅	**non-cancerous**	*<5%*	❌	❌	❌	❌
** *11, 12, 13,* ** ** *14, 15* **	—	✅	✅	**normal liver**	*<5%*	❌	❌	❌	❌

**Table 2 toxins-18-00014-t002:** Immunohistochemical expression patterns, cutoff values, and interpretative criteria for lineage assignment and pathway status. This table summarizes the subcellular localization of immunoreactivity for each protein, the predefined cutoff values used to classify protein expression as positive, and the corresponding biological interpretation.

Protein	Expression Pattern	Cutoff Value for Positivity	Interpretation
**Cytokeratin 19**	Membranous/ cytoplasmic	≥10% of positive tumor cells	Cholangiocellular lineage
**Hep Par-1**	Cytoplasmic	≥10% of positive tumor cells	Hepatocellular lineage
**p53**	Nuclear	≥10% of tumor cell nuclei positive	Nuclear p53 accumulation
**Cyclin D1**	Nuclear	≥10% of tumor cell nuclei positive	Nuclear cyclin D1 overexpression
**Rb**	Nuclear	Complete absence of nuclear staining in tumor cells	Loss of Rb protein expression
**β-catenin**	Nuclear	≥10% of tumor cell nuclei positive (with/without cytoplasmic shift)	Aberrant activation of Wnt/β-catenin pathway

## Data Availability

The original contributions of this study are included in the article. Further inquiries can be directed to the corresponding author(s).

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
