# Peer review of "Aflatoxin Mixture-Driven Intrahepatic Cholangiocarcinoma in Rats Involving G1/S Checkpoint Dysregulation"

_toxins, 2025, doi:10.3390/toxins18010014_

Round 1

Reviewer 1 Report

Comments and Suggestions for Authors

1. The exposed group includes only 10 rats, with 8 developing hepatic tumors and 12 tumors in total. The control group (n = 5) is relatively small and apparently free of lesions. While the descriptive nature is acknowledged, the current wording sometimes overstates the strength of the evidence (e.g., “establishing a robust iCCA model” and “clearly indicate”). Provide exact iCCA incidence per group (animals with ≥1 iCCA / total per group), and clearly distinguish animal-level vs tumor-level denominators throughout. Temper causal language in the abstract, discussion, and conclusions. The data support that this exposure can induce iCCA-phenotype tumors under the described conditions, but the small cohort and lack of dose–response preclude strong generalization.
2. The manuscript states that analyses were descriptive and no formal hypothesis testing was performed. However, terms like “frequent”, “substantive subset”, and “predominant” are used without any measure of variability or uncertainty. Provide basic descriptive statistics for key endpoints (e.g., proportion of positive tumors with 95% CI, where feasible, or at least explicit denominators in the main text and/or a table). Clearly justify the absence of any inferential statistics (e.g., due to exploratory design) and explicitly mention this limitation in the Discussion.
3. The dose is described as “equivalent to 400 µg AFB1 per kg of diet”, but the real-world relevance and relation to known aflatoxin exposure scenarios remain underdeveloped. Explain briefly how this dose relates to established experimental hepatocarcinogenesis models and whether it approximates high-end human exposure scenarios or is supra-physiological. Clarify how the dose equivalence to “per kg of diet” was calculated from the gavage mixture and animal body weight / food intake assumptions. A short calculation or reference would improve transparency.
4. The text focuses on the 12 iCCA lesions, and briefly mentions one cHCC-CCA. It is not entirely clear whether all 12 tumors are pure iCCA except the single mixed lesion, or whether other patterns were observed and then reclassified as iCCA based on IHC. Provide a concise table listing each tumor with its final classification (iCCA vs cHCC-CCA), morphology (architecture), and lineage marker profile (CK19, Hep Par-1). Clarify in the Results how many tumors were unequivocally iCCA by morphology alone vs those requiring IHC to resolve the lineage.
5. The manuscript makes strong mechanistic inferences (e.g., “toxin-driven iCCA axis centered on TP53 pathway disruption and accelerated G1→S transition”, “convergent mechanisms of G1/S deregulation”) based solely on IHC of p53, cyclin D1, and Rb without any genetic or functional assays. Soften mechanistic language and clearly frame these as inferred mechanisms consistent with IHC surrogates rather than demonstrated pathways. Add an explicit paragraph in the Discussion noting that p53 accumulation is not synonymous with TP53 mutation, and that Rb “loss” by IHC does not discriminate among mutation, epigenetic silencing, or functional inactivation. This is touched upon but should be more prominent and linked directly to the strength of the conclusions.
6. In PCNA and proliferation data presentation, the text reports a mean PCNA index of ~69% (range 50–80%), but the data are not shown in a structured form. Present the PCNA proliferation index for each tumor in a table or figure (box plot or bar graph) with individual values. If possible, compare PCNA labeling between tumor tissue and non-tumorous liver in a concise visual way. 
7. While comparison with existing aflatoxin-induced cholangiocarcinoma models, previous hamster and primate data are cited, the comparison could be more explicit. Add a short paragraph in the Discussion summarizing how your rat model compares to prior aflatoxin-associated cholangiocarcinoma reports in terms of species, dose, latency, lesion type (iCCA vs cHCC-CCA), and molecular markers. This will help position the novelty more clearly.
8. The manuscript suggests the CDK4/6 axis and p53 reactivation as potential therapeutic avenues and refers to “actionable biological nodes”. While interesting, these statements may be too far-reaching given that no intervention or pharmacologic testing has been performed. Rephrase these sections to emphasize that these are hypotheses generated from the IHC profile, and clearly separate speculative future directions from data-supported conclusions.
9. Consider including the number of animals per group explicitly in the Abstract (10 exposed, 5 controls) and specify that analyses were descriptive, without formal statistical testing. 
10. The phrase “Subchronic exposure… yields an iCCA model” could be softened to “can yield” or “resulted in”. 
11. Ensure that all percentages (e.g., CK19 100%, p53 83%, cyclin D1 67%, Rb loss 42%, nuclear β-catenin 8%) are consistently accompanied by numerators and denominators (e.g., 10/12, 8/12). This will aid readers and avoid confusion.
12. Figures 1–3 are informative, but the legends could more clearly state the stain and magnification for each panel and, where relevant, whether the image is representative of the majority of tumors or a particular subset (e.g., Rb-loss vs Rb-retained tumors). 
13. If possible, add at least one panel illustrating the mixed cHCC-CCA case with both components and their differential marker expression.
13. Under materials and methods, indicate the approximate body weight range at the start and end of exposure. Clarify whether any animals were euthanized before the 12-month endpoint due to humane endpoints and whether their tumors were included in the total count. Please specify if pathologists were blinded to group (exposed vs control) during histologic and IHC evaluation (the text suggests blinding, but this should be clearly stated for all assessments). The ARRIVE 2.0 compliance statement is appreciated. You may briefly indicate whether a prior sample size justification was attempted, and if not, state this explicitly as a limitation.
13. Some DOIs appear duplicated or repeated at the end of the text block. Check for redundant DOI lines and ensure that all references follow the journal’s citation style. Verify that abbreviations and gene names (TP53, CTNNB1, IDH1/2, etc.) are formatted consistently.

Comments on the Quality of English Language

Overall English is good, but a light language edit would help smooth a few long sentences and ensure consistent tense (especially when switching between describing results and interpreting them). Terms like “clear evidence” or “concrete leads” may be perceived as too strong for a small exploratory study. Consider more neutral wording (e.g., “supportive evidence”, “suggestive leads”). Ensure consistent use of “subchronic” vs “chronic”, and “aflatoxin mixture” vs “AFB1-dominant mixture”.

Reviewer 2 Report

Comments and Suggestions for Authors

This manuscropt describes the effect of oral administration of afratoxin on mice. Subchronic oral exposure to an AFB1-dominant aflatoxin mixture induced intrahepatic cholangiocarcinoma (iCCA) in male Wistar rats. Histology revealed glandular/tubular architecture, cytokeratin-19 positivity, and high proliferative activity (PCNA of 69%). Frequent p53 accumulation (83%) and cyclin D1 overexpression (67%) indicated G1/S checkpoint disruption, while Wnt activation was rare. Findings expand aflatoxin’s oncogenic spectrum and warrant genomic signature confirmation.

The research is interesting, but there are several drawbacks in this manuscript.

methods lack clarity and details. authors should write subtitles for each experiments. Concrete manufactures and concentrations are lacking.

description of figures lack clarity. it is hard to know which panel represents for what. 200 and 400 seem similar and either is fine. in the text, they should describe which result is describing which panel.

Round 2

Reviewer 1 Report

Comments and Suggestions for Authors

Nil

Author Response

We thank you for considering this study ready for publication.

Reviewer 2 Report

Comments and Suggestions for Authors

The authors have improved the manuscript, but it is not sufficient.

For example, how they quantified the proteins using antibodies are not well written.

Author Response

We would like to thank the reviewer again for their comments. We believe that all concerns have been adequately addressed in the revised version of the manuscript.

The authors have improved the manuscript, but it is not sufficient.

For example, how they quantified the proteins using antibodies are not well written.

Answer: Protein expression quantification was performed semi-quantitatively using immunohistochemical analysis. The description of the methodology and interpretation of the results has been improved. Please see pages 12 and 13, lines 383 to 446, and Table 2.